

**A Wavelength Dispersive Instrument for Characterizing Fluorescence and**
**Scattering Spectra of Individual Aerosol Particles on a Substrate**
Donald R. Huffman[1], Benjamin E. Swanson[2], and J. Alex Huffman[2*]
[1] University of Arizona, Department of Physics, Tucson, Arizona
[2] University of Denver, Department of Chemistry and Biochemistry, Denver, Colorado
*Correspondence to*: J. Alex Huffman (alex.huffman@du.edu)
**Abstract:** We describe a novel, low-cost instrument to acquire both elastic and inelastic
(fluorescent) scattering spectra from individual micron-size particles in a multi-particle
collection on a microscope slide. The principle of the device is based on a slitless spectroscope
often employed in astronomy to determine the spectra of individual stars in a star cluster, but that
had not been applied to atmospheric particles. Under excitation, most commonly by either a 405
nm diode laser or a UV light emitting diode (LED), fluorescent emission spectra of many
individual particles can be determined simultaneously. The instrument can also acquire elastic
scattering spectra from particles illuminated by a white light source. Advantages and
disadvantages of using black-and-white cameras compared to color cameras are given. The
primary motivation for this work has been to develop an inexpensive technique to characterize
fluorescent biological aerosol particles. An example of an iPhone-enabled device is also shown
as a means for collecting data on biological aerosols at lower cost or utilizing citizen scientists
for expanded data collection.
1. **Introduction**
Primary biological aerosol particles (PBAP) suspended in the atmosphere, often termed
bioaerosols, are comprised of a complex mixture of biological organisms and materials including
bacteria, fungal spores, pollen, and fragments and excretions of plants, animals, and
microorganisms (Després et al., 2012). They can influence human, animal, and agricultural
health by causing disease and triggering allergies, and can influence Earth systems such as cloud
formation and the hydrological cycle by acting as nuclei for the formation of liquid water or ice
cloud droplets (Douwes et al., 2003; Möhler et al., 2007; Morris et al., 2014; Pöschl et al., 2010;
Pöschl and Shiraiwa, 2015). Yet large holes in the collective understanding about atmospheric
bioaerosols remain, and PBAP have become increasingly important in recent years in aerosol
science due to synergistic advancements in detection technologies and in the understanding of
roles these aerosols may play in the atmospheric system.

One common method for the discrimination of biological particles from the rest of atmospheric
particulate matter is the use of single-particle laser-induced fluorescence (LIF). Many research
groups have utilized LIF for real-time investigation of bioaerosols, often for the purpose of
detecting possible agents of bioterrorism (e.g. Després et al., 2012; Kiselev et al., 2011;
Manninen et al., 2008; Pan et al., 2003; Pan et al., 2010; Saari et al., 2014; Sivaprakasam et al.,
2004; Sivaprakasam et al., 2009). Over the last 10-15 years a number of LIF instruments have
also become commercially available and are becoming more widely used for atmospheric
research. Two of the bioaerosol-LIF instruments most commonly used within the atmospheric



research community are the ultraviolet aerodynamic particle sizer (UV-APS; TSI, Inc.,
Shoreview, MN) (Hairston et al., 1997; Huffman et al., 2010) and the wideband integrated
bioaerosol sizer (WIBS; DMT, Inc., Boulder, CO) (Foot et al., 2008; Gabey et al., 2010; Kaye et
al., 2005; Perring et al., 2015). Both instruments allow characterization of particles with high
time- and size-resolution, but offer little spectral discrimination. The UV-APS provides
aerodynamic particle size-resolved information about ensembles of particles averaged for several
seconds or minutes, and fluorescence excited at 355 nm is provided as the sample-average
emission intensity in one wavelength band between 420 – 575 nm (Hairston et al., 1997). The
WIBS provides optical particle size and fluorescence information for each particle interrogated,
using two excitation wavelengths of 280 nm and 370 nm, and detects fluorescence emission in 2
channels for each excitation (Foot et al., 2008). Further, both instruments are quite expensive,
selling for approx. $100k or more. The poor spectral resolution limits the ability of these
instruments to finely discriminate between particle types, and the high cost of these instruments
prevents their widespread global application.
If instruments of considerably less cost and smaller size could be developed and distributed with
the capability to discriminate biological particles from non-biological material, the information
on the global properties of biological aerosol particles and their effect on human health and
climate could be collected on a much more comprehensive scale. We have developed a low-cost
instrument capable of characterizing fluorescence and white-light spectra from many individual
particles collected onto a substrate, each excited at several excitation wavelengths. Here we
introduce both a benchtop and a portable, smartphone-based instrument that each may serve as a
transformative tool for bioaerosol detection with application to atmospheric research, pollen
monitoring, and mold spore detection, among many other applications. The instrument design
also has potential application to any scientific or medical problem where a minority of micron-
sized particles can be discriminated from a majority of particles based on their fluorescence or
elastic scattering spectra.
**2.  Instrument description**
The instrument is built around a conventional compound microscope and uses standard glass
microscope slides as particle substrates, though any non-fluorescent and relatively non-reflective
material may be utilized. We will explain the principle and the details with reference to a three-
stage progression of the idea as shown in Figure 1. The optical principle is derived from the
simple spectroscope as shown in Figure 1a. In this venerable instrument, the light to be spectrally
analyzed is directed onto a narrow incident slit. The light from this slit is made parallel by the
collimating lens and impinges on a wavelength-dispersive element such as a prism or a
transmission grating (as shown). Parallel light from the grating is dispersed into an angle $\Theta$ by
the grating according to the grating equation for normal incidence (Jenkins and White, 1957),

$$d \sin \theta = n\lambda \qquad (1)$$

where $d$ is the distance between rulings on the grating, $n$ is an integer giving the order of
diffraction, and $\lambda$ is the wavelength of the light. For the example in Figure 1a an atomic emission
source such as H, He, or Hg is shown as the light source, which illuminates a narrow slit from
behind. An image of the slit at each wavelength is dispersed according to the angle defined in
Equation 1. Because there are multiple wavelengths emitted in the visible region by the sources
just mentioned, there will be a real image of the slit, which gives rise to the "line spectrum" that



can be viewed with an eyepiece. This real image can also be captured on photographic film or on
an array detector such as a charge-coupled device (CCD) or complementary metal-oxide
semiconductor (CMOS) detector in a digital camera.
The essential innovation of the present instrument is understood by imagining the replacement of
the back-illuminated slit by one or more particles in place of the slit, held on the plane of a
microscope slide as in Figure 1b. For detection at a $\Theta$ angle of zero (viewing straight through),
the dark-field image appears simply as a collection of illuminated particles, and no spectral
information is gained. Viewed at an appropriate angle through the transmission grating, however,
the individual particles will exhibit spectral dispersion. For monochromatic illumination of a
given particle, such as by a single spectral line or a laser beam, the first order diffraction will be
a single spot imaged in the same plane that the slit image was shown previously. In the version
shown in Figure 1b, however, the image at the detector will be located at vertical and horizontal
coordinates corresponding to an image of the particle's position on the substrate. For white light
illumination the dispersed image of each particle will be a swath of light with a height
corresponding to the diameter of the particle and a length corresponding to the difference in
dispersion angle for visible wavelengths. Each particle will display the various rainbows of
colors when viewed by eye through an eyepiece. If the substrate consists of a number of particles
distributed sparsely on the surface of a microscope slide as shown in Figure 1b, each particle will
be imaged as either a single, colored point for illumination by a monochromatic source, a series
of different colored points for a multiple line emission source, or a continuum spectrum for white
light illumination.
The main idea here is the replacement of an entrance slit, which is present in most spectroscopes
and the spectrographs, monochromators, and spectrometers built upon the spectroscope idea of
Figure 1a, with individual small particles each of which effectively act as very small (two-
dimensional) slits as in Figure 1b. Thus this instrument falls into the category of a slitless
spectrometer, which has been used widely, but mostly in astronomical studies. In the 1880's
Edward Pickering implemented the idea by placing a large prism in front of the objective lens of
a refracting telescope and recording the spectral swaths on photographic film for analysis (Hale
and Wadsworth, 1896). In his classic textbook George Abell (1969) has shown images of the
color spectra from all stars in the famous Pleiades star cluster in one image. With this as
inspiration, we produced the spectral streaks from microscopic particles that look very much like
the streaks from stars through a telescope.
The incarnation of the instrument described here is relatively simple, yet has not been applied for
aerosol particle research. Other commonly applied techniques provide aspects of the benefits
described here, but are yet distinctly different (Lakowicz, 2010). Fluorescence microscopy is
widely applied in many scientific fields, but most commercial fluorescence microscopes provide
fluorescence information by utilizing relatively broad-wavelength filters for excitation and
emission. As a result, micrographs can be achieved that utilize a given pair of excitation and
emission windows, but spectral information can only be attained by using a large sequence of
filters. Laser-scanning confocal microscopy allows the user to excite a given spot with a chosen
laser source and then measure the spectral dependence of the fluorescent emission, but scanning
a collection of particles can be time-consuming, because the laser must raster-scan through the
entire field of view, and the instrumentation is costly and relatively complicated. Fluorescence



spectroscopy is achieved in the bulk phase, most often for solutions and suspensions, but is also
possible for powders and solid materials, by measuring fluorescence emission spectra as the
excitation wavelength is varied. In this case, highly resolved fluorescence emission spectra can
be acquired, though fine spatial resolution is typically not possible and thus the technique
typically cannot provide single particle spectra. Single-particle fluorescence instruments, such as
are used for detection of atmospheric bioaerosols, normally provide very poor spectral resolution
(total of 1-3 emission channels) and are very expensive (Foot et al., 2008; Hairston et al., 1997;
Manninen et al., 2008; Sivaprakasam et al., 2004). A notable exception to the statement about
spectral resolution is the single particle fluorescence spectrometer (SPFS) developed by the
Army Research Laboratory (Pan et al., 2011; Pan et al., 2010). This instrument can provide
highly resolved spectra of individual aerosol particles suspended in air flowing through the
device, however, the instrument is one-of-a-kind and is very expensive ($100k's). Thus, the
instrument described here combines benefits of each of these other instrument concepts by
delivering spectra of each particle in a collection on a substrate, each at a number of excitation
wavelengths, and at a cost orders of magnitude lower than other techniques. Recently a similar
concept of slitless microscope spectroscopy was used with silver nanoprobes applied to in vivo
monitoring (Cheng et al., 2010; Xiong et al., 2013). This appears to have been the first
application of slitless spectroscopy to single particles, however, the application was towards
biomedical testing with no application to atmospheric aerosols and not towards a portable device.
*2.1 Typical operation*
Figure 1c adds some practical details about the instrument described here, the original of which
was constructed from a student microscope (Model 656/98, SWIFT Microscopy, Carlsbad, CA)
formerly used in an undergraduate biology teaching lab. The original vertical microscope was
placed in a horizontal orientation, with the sample slide illuminated by one of several light
sources consisting of diode lasers, LEDs, or a tungsten filament light bulb. Illumination of the
particles from above and to the side of the stage produces a bright particle image on a dark
background - one example of so-called dark field illumination. The microscope stage with its
attached x-y positioner and the objective lens turret were retained from the original microscope.
The eyepiece was removed and placed on a rail that rotates about a pivot point located in the
plane of the transmission diffraction grating (300 grooves/mm; Thor Labs, Inc., GT25-03). A
standard optical table with mounting holes spaced on a one inch grid supports optical rails and
holders secured to it for mounting the optical components (Thor Labs, Inc.). Various cameras,
both color and black and white (b/w) have been adapted with mounts to the rotating detector
arm.
In operation, when the camera arm is set at zero degrees, the instrument operates as a standard
microscope. At the approximate angle of first order diffraction the camera will record spectral
swaths of each particle in the field of view. We note that either elastic scattering (no change in
wavelength upon scattering) or inelastic scattering (change of wavelength upon scattering) can
be recorded. In all cases discussed in this paper inelastic scattering is due to fluorescence, though
the concept will apply to inelastic Raman scattering, for example, and this will be investigated in
future research. Thus the instrument has the capability of recording either elastic or inelastic
scattering from individual particles in its field of view (of the order of one mm), depending on
the type of particle and the illumination source. As a practical matter, the density of particles
distributed on the slide should be sufficiently sparse that the spectral swaths do not overlap if
individual particle spectra are to be determined. An additional optical element necessary for



fluorescence spectroscopy is a long-pass blocking filter which we place between the objective
lens and the grating, selected so as to block the emission of the fluorescence-exciting source, but
transmit the fluorescence spectrum to a wavelength as close as possible to the blocked laser
wavelength. For example, we commonly use a diode laser with a wavelength of about 405 nm
and long-pass filters chosen to block wavelengths below approx. 420 or 440 nm *(*Edmund
Optics, 832916-10*).* We have also used 280, 370, and 450 nm excitation sources, but the process
and concept remains the same.

### 189  *2.2 Example spectra and data analysis*

Individual panels of Figure 2 (a through d) serve to illustrate the different types of images
collected during standard operation. Each panel of Figure 2 shows a sample of paper mulberry
pollen (*Broussonetia payrifera*; 12-13 µm; Allergon, SKU 0578) collected onto a conventional
microscope slide after aerosolization by blowing air into a tube of pollen. Figure 2a shows pollen
grains illuminated with red laser light and imaged as a standard micrograph, with the camera arm
at zero degrees. Figure 2b is taken with illumination by the red and violet lasers simultaneously
(wavelengths 650 nm and 405 nm, respectively). Dots on the left and right of each streak
correspond to monochromatic images of the particles illuminated by the red and violet lasers,
respectively. These appear as washed-out dots devoid of much color and large in size, as a result
of the intensity of the illuminating lasers which saturate the detector. The swaths of color
between the dots are due to fluorescence induced from the pollen particles. It should be noted
that each of the spectral streaks in Figures 2b through 2d can be related to specific particles by
way of their images in Figure 2a. Figure 2c shows an example white-light scattering spectra after
illumination by a polychromatic tungsten filament. Figure 2d shows an identical image to that of
Figure 2b, but without the red laser and with the blocking filter in place. In this way, the
fluorescence spectra can be detected more sensitively, without wash-out from the red and violet
lasers. By comparing Figures 2c and 2d one can see that the relative fraction of pollen particles
fluorescent in this sample is nearly 100%, since this particulate sample is made up of a single
kind of pollen.
Two laser spots of known wavelength give the ability to calibrate the pixel location with respect
to wavelength for each of the spectral swaths. The wavelength scale for each particle is
established by determining the wavelength dispersion from the positions in pixel numbers of the
red and violet calibration spots, assuming a linear dependence of wavelength on distance in the
plane of focus. Making use of the open source program suite Image J (Rasband, 1997) one draws
a box in the region of interest closely around the spectral swath. Employing the program's ability
to determine average intensity at each horizontal pixel location on the swath, the data is
presented in two columns – one column corresponding to horizontal pixel number and the other
column corresponding to average light intensity detected at the given pixel number. These data
columns are entered into a spread sheet and the pixel numbers are converted to a column of
wavelength values using the dispersion calibration already determined.
Figure 3 shows the normalized fluorescence spectrum of three particles from the fluorescence
image in Figure 2d acquired by determining the intensity as a function of wavelength, as
discussed above. The result is a broad peak centered at approx. 500 nm, broadly consistent with
previous literature for many types of pollen (Hill et al., 2009; Pan et al., 2011; Pöhlker et al.,
2013; Pöhlker et al., 2012). A secondary peak, caused by the color camera pixels centered at
approx. 600 nm, is also clear and will be discussed in a following section.



## 3. Applications


In Figure 4 we present images for two different collections of particles that highlight the simplest
application of the instrument, which is to estimate the fraction of particles in a mixture that
exhibit fluorescence. This can be achieved irrespective of whether emission spectra are
measured, and can have important application to a number of scientific questions. For example,
the UV-APS has been frequently applied to the detection of bioaerosols in ambient air (Huffman
et al., 2010), but it cannot provide spectrally resolved information and thus essentially only
provides information about the fraction of particles that exhibit fluorescence at a given set of
excitation and emission wavelengths. Figure 4 shows images associated with particulate samples
of polydisperse particle size, as opposed to those shown in Figure 2 which are very uniform in
size.

### 3.1 Outdoor sample particles

The particles shown in Figures 4a-c (top panel) were collected by leaving a glass microscope
slide lying horizontally on a table three feet above ground level outdoors in Tucson, Arizona for
24 hours in the winter time in clear weather. The comparison of the fluorescence image (Fig. 4c)
with the elastic scattering image (Fig. 4b) is an illustration that the fraction of aerosol particles
exhibiting fluorescence is relatively easily achieved without needing to analyze spectra and thus
with little image processing. In fact, a rough idea about the relative fluorescent fraction can be
achieved by eye simply by looking at camera images using both illumination schemes (e.g. Figs.
4b-c). The count for elastic scattering particles, which should comprise all particles both
fluorescent and non-fluorescent, is approx. 46 in this example, whereas the number count of the
brightly fluorescent particles is approx. 7. Thus the fraction of total particles that are fluorescent,
and presumably of biological origin, is approx. 7/46 here or approx. 9%, which is broadly
consistent with typical ambient measurements, though fluorescent biological aerosol
concentrations and properties vary widely by season and geography (Huffman et al., 2013;
Schumacher et al., 2013). This fraction is also highly dependent on the threshold one applies to
categorize a given particle as fluorescent or not, and fluorescence intensity is strongly a function
of particle size. Most fluorescence-based aerosol detectors are faced with this difficult challenge
of categorization, and the sensitivity of a given detector will significantly influence the
comparison of the relative fraction of fluorescent particles detected by any two instruments or
types of instruments (Healy et al., 2014; Hernandez et al., 2016; Huffman et al., 2012; Saari et
al., 2013).

### 3.2 Fused silica particles

For the purpose of illustrating results for a particle collection expected to show no fluorescence
we prepared a collection of particles by grinding a piece of optical fused silica with an agate
mortar and pestle. The optical silica of the starting material shows no appreciable fluorescence in
bulk throughout the visible spectral region. A large spread of particle sizes is apparent in the
micrograph of Figure 4d with many overlapping elastic scattering spectra, apparent under
illumination with the white light of a tungsten lamp (Fig. 4e). Figure 4f, however, shows one
unexpected, strongly fluorescent particle and approximately three other weakly fluorescent
particles. The elastic scattering (Fig. 4e) for these particles is not particularly remarkable. Our
interpretation is that the very few fluorescent particles observed are contaminants of fluorescent
material, which entered into the mixture during grinding or handling in a relatively dirty
laboratory, and which had previously contained various pollen particles. We refer to the image



series in Figure 4d-f as "a fluorescent needle in an elastic scattering haystack." It illustrates how
biological impurities might easily be detected with our apparatus even in the presence of a large
majority of non-biological, or otherwise non-fluorescent particles.
**4.   Comparison of black-and-white with color cameras**
Results presented to this point have been based on images taken using a Canon Powershot
A2300 HD color camera, which is a simple point-and-shoot digital camera costing less than
$100. It was chosen to promote the goal of producing a relatively low cost apparatus. We have
also used a more sophisticated black and white (b/w) or monochrome camera – a Luminara
Infinity 2-1R costing about $2000. Color cameras in general have several disadvantages
compared to b/w cameras, as well as some advantages. They use CCD or CMOS detector arrays
with three different color pixels and filters having peak sensitivities in (for example) the red,
green, and blue. When a broad continuum of light is incident on such a detector, as from a
tungsten lamp or a broadband fluorescence spectrum from a pollen grain, the otherwise smooth
spectral curve will show variations due to the color pixel characteristics. Figure 5 shows this
effect experimentally with the white light spectrum from an incandescent tungsten-filament bulb
(General Electric, Miniature Lamp 210, B6, 6.5V) scattered from ground particles of sodium
chloride (NaCl; Sigma Aldrich, CAS# 7647-14-5) and independently detected by the
monochrome camera and color camera discussed above. Each NaCl particle is expected to
exhibit reasonably smooth scattering *vs*. wavelength, because its large size relative to the
wavelength and its irregular shape do not promote the various sphere-like resonances expected
from Mie theory (Bohren and Huffman, 1983). The monochrome camera yields a relatively
smooth, broad scattering curve. For comparison, Figure 5 also shows the emission spectrum from
a 3000 Kelvin blackbody multiplied by the theoretical sensitivity curve of the CCD used in the
monochrome camera[1]. The theoretical blackbody curve and the measured elastic scattering curve
match closely, suggesting that the monochrome camera introduces very little aberration as a
function of wavelength. In contrast, the color camera shows a spectrum with pronounced peaks
that are due to the different color pixels. These strong variations in the spectra from color
cameras are difficult to correct for and are not present in monochrome cameras. Another
disadvantage of most color cameras is the incorporation of an infrared blocking filter which is
added to exclude light of wavelengths longer than deep red in order to produce a more natural
color in the image, but which limits the red spectral range of fluorescence and elastic scattering
detection.
To further illustrate both of these undesirable features of color cameras, Figure 6 shows a
comparison of images from particles of Kentucky bluegrass pollen (*Poa praetensis, also known*
*as smooth or common meadow-grass;* 20 - 120 μm; Allergon SKU 0116), recorded and analyzed
as described above to derive the spectral curves from both the b/w (black curve) and the color
camera (red curve). The b/w camera results show a broad fluorescence band peaking near 490
nm and a somewhat narrower band near 680 nm. The latter band is assigned to chlorophyll-a,
which is present in most grass pollens (Maxwell and Johnson, 2000; O'Connor et al., 2011;
O'Connor et al., 2014). A reference spectrum of bulk bluegrass pollen powder was achieved by
adding approx. 5 mg of pollen to one well of a black 96-well plate (Fisher Scientific, 07-200-
329) and recording a fluorescence emission spectrum at 405 nm excitation using a microplate
reader spectrofluorometer (Infinite M1000 Pro, Tecan, Männerdorf, Switzerland). This technique

---

[1] http://www.opticstar-ccd.com/Images/Astronomy/Imagers/OS/Common/QE-ICX205AL-594x255.jpg





cannot provide single-particle spectra, as discussed previously, but delivers a spectrum as an
average of the bulk powder. The peak locations in the reference and b/w camera spectra are
identical, though the relative ratio of chlorophyll peak to main peak is higher in the reference
spectrum. This is expected, as individual pollen grains exhibit markedly different concentrations
of chlorophyll as a result of differences in age and physiological state (Boyain-Goitia et al.,
2003; Pöhlker et al., 2013). The color camera also shows a band near 490 nm as well as an
irregular and asymmetric peak near 600 nm, with no sign of the chlorophyll band. The ~600 nm
structure appears in the color camera image due to the color pixel effect discussed above. The
absence of the chlorophyll peak from the color camera is likely due to the presence of the
infrared blocking filter in the color camera only, which removes the transmission of chlorophyll
transmission.
Notwithstanding the disadvantages of the color camera, it has several pleasant and useful
features. First of all, the images of the spectra are simply interesting and beautiful! This may
even be a non-trivial benefit when soliciting effort from citizen scientists, because the images
can produce captivating, artistic views of the natural, microscopic world. From a practical
scientific viewpoint, the colors are valuable for quickly getting oriented to the approximate
wavelength positions of spectral features, which may be evident in the spectral swaths even
without further processing. Also, the spectral colors are very useful when higher order spectra ($n$
> 1 in Equation 1) are present, which may result in overlapping orders. These can be easily
sorted out if colored swaths are present, while they can become confusing when viewing the
colorless black and gray images from a monochrome camera. For many investigations we have
utilized both cameras in tandem, and perhaps the best solution in the future will be to use a
combination of a b/w camera along with a color camera for standard usage. The two could be
arranged on optical axes at 90 degrees from one another with a "flip mirror" used as a quick
method to switch from one to the other.
**5. Smartphone embodiment**
Because of the development of smartphones for the vast, global consumer market, smartphones
now contain highly sophisticated cameras built into the devices which are light-weight, low in
electrical power, and relatively inexpensive. These devices also can have other useful capabilities
such as GPS sensors providing geographic location, temperature and time detection, and the
capability of connecting easily to the internet for almost immediate sharing of data. We built and
tested two prototypes of the particle spectrometer – one for an iPhone and one for an Android
phone. Figure 7a shows a photograph of the iPhone version, with an iPhone model 5s (Apple,
Inc.) placed on top. The iPhone uses its own battery for the camera and other smartphone
functions. Behind the wooden panels of the instrument body (13.3 x 13.3 x 7.4 cm; 58 g) are the
same optical components discussed above regarding the benchtop version. There is an objective
lens, blocking filter, and diffraction grating as well as three light sources (red and violet laser
diodes and tungsten light source) operated by 2 AA batteries. The microscope slide onto which
particles are collected can be slipped through a hole in the exterior casing, and four switches on
the side of the instrument, attached to the anodized aluminum panel, operate the light sources so
that images can be acquired without opening the case to ambient light. At present the little
instrument only acquires and saves images, which must be downloaded to a computer for
processing, as described earlier in this paper. A future goal is to utilize a combination of on-
board and cloud-based image processing to provide spectra to the smartphone user.





The second panel (Figure 7b) shows a typical image of Kentucky bluegrass pollen taken with
this instrument. The image shows fluorescence streaks from individual particles, similar to
images acquired using the benchtop instrument (Fig. 2) as discussed above. Because of the wide
angle of acceptance of the iPhone camera, the $\Theta = 0$ microscope image is visible in the middle of
the image as well as first order diffraction streaks to both the left and right. The fluorescence
intensity of the right-most image is less than for the light diffracted to the left due to the blaze
angle of the grating.  The peak at ~600 nm arises from the wavelength dependence of the color
pixels, as discussed above. It is important to note here, however, that the peak at ~680 nm shows
the chlorophyll-a peak, in contrast to the absence of this peak when using the Canon color
camera discussed previously.

**6.  A vision for broad scale use**

Our vision for broad-scale use of a portable version of the instruments is twofold. First, the
ability to sample and analyze fluorescence spectra of airborne particles has been limited because
UV-LIF instruments are so expensive that they are typically deployed one at a time. This means
that only in rare cases have such instruments been utilized to record information about
bioaerosols to understand spatial variability. With the development of a sampler for particle
collection hyphenated to the optical analysis tool described above, it could be possible for many
units to be constructed inexpensively and deployed simultaneously, thus allowing for the
collection and analysis of particles in a network across a chosen area. For this to become a reality
we are working towards a future model which will incorporate automatic sampling and analysis
capabilities, for example utilizing a Raspberry Pi camera to reduce the need for a dedicated
smartphone for each unit.
A second vision, specifically applicable to the smartphone version, is to involve interested
citizen scientists from around the world who already possess the most expensive components of
a fluorescent instrument – the smartphone camera. Our opto-mechanical additions to the smart
phone meet the desired requirements of being relatively light weight, low power and
inexpensive. At an estimated cost in bulk of about $200, some 500 units (not counting the
smartphone) could perhaps be produced for the approximately $100,000 price of a real-time,
commercial sensor such as the WIBS. Interested persons could be enlisted to collect and measure
particles and send the results back to a central computer, where analysis would be done. Some
simplified results, such as the percentage of fluorescent particles of biological origin, might be
returned to the volunteer measurer to stimulate and maintain their interest. Or, a cloud-computer
may allow spectra from individual particles to be clustered and compared with a database of
spectral standards for course-level discrimination (Pan et al., 2012; Pinnick et al., 2013;
Robinson et al., 2013). Citizen scientist-assisted collection of data about fluorescent aerosol
particles using this technique could help change the face of this area of science by acquiring
orders of magnitude more data points in time and space than are currently available.
Although this paper has emphasized the applications of the instrument for acquiring spectral
fluorescence of particles, there may occur even more applications for elastic scattering from
particles, which can be obtained using white light illumination as in Figure 2c rather than 2d. The
technique of acquiring spectra from individual particles can perhaps also be applied to the
acquisition of Raman scattering spectra, though this will introduce additional technical
challenges. Recently an instrument for real-time detection of single particles in air by Raman
spectroscopy has been made commercially available (Hill et al., 2015; Ronningen et al., 2014).



The instrument described here could be developed in the future to provide Raman spectroscopy of individual atmospheric particles, with reduced resolution or signal-to-noise, but also with significantly reduced cost.

## 7. Summary and conclusions

We have described the development of a small, light-weight and low-cost instrument which uses the principle of a slitless spectrograph to determine both the elastic scattering spectra and inelastic spectra (such as fluorescence) for each particle in a many-particle collection on a glass microscope slide or other surface. In addition to a benchtop model composed of standard microscope parts, we have shown a small instrument as an attachment to a smartphone or other small digital camera producing data in the form of images that can be sent immediately from almost any location on the earth to a remote, master computer for analysis. In the case of its primary intended use, the instrument can provide separate spectral images of fluorescence and elastic scatter with which a simple count on these two images is sufficient to determine the ratio of biological particles to total particles. At present, there are no inexpensive, autonomous sensors available that can estimate the concentration of mold spores or pollen types that can exacerbate human allergies. The portable version of the instrument may provide a critical leap in the detection of several types of biological aerosol particles. For example, by differentiating between chlorophyll-containing pollen and from other pollen types, the detector could provide a quick quantification of grass-type pollens (i.e. *Ambrosia* or ragweed) that are responsible for many cases of hay fever and allergenic rhinitis across the world (D'Amato et al., 2007; O'Connor et al., 2014).

**Acknowledgements**

The authors acknowledge Brason Holt for assistance with laboratory experiments. DRH acknowledges the $C_{60}$ Patent Royalty Fund of the University of Arizona for support. JAH and BJS acknowledge funding from the University of Denver professional research opportunities for faculty (PROF) program.



**References and links**
Abell, G. O.: Exploration of the Universe, Holt,Rinehart & Winston of Canada Ltd, 1969.
Bohren, C. F. and Huffman, D. R.: Absorption and Scattering of Light By Small Particles, John
Wiley & Sons, New York, NY, 1983.
Boyain-Goitia, A. R., Beddows, D. C. S., Griffiths, B. C., and Telle, H. H.: Single-pollen
analysis by laser-induced breakdown spectroscopy and Raman microscopy, Applied Optics, 42,
441 6119-6132, 2003.

Cheng, J., Liu, Y., Cheng, X., He, Y., and Yeung, E. S.: Real Time Observation of Chemical
Reactions of Individual Metal Nanoparticles with High-Throughput Single Molecule Spectral
Microscopy, Anal. Chem., 82, 8744-8749, 2010.
D'Amato, G., Cecchi, L., Bonini, S., Nunes, C., Annesi-Maesano, I., Behrendt, H., Liccardi, G.,
Popov, T., and van Cauwenberge, P.: Allergenic pollen and pollen allergy in Europe, Allergy,
447 62, 976-990, 2007.

Després, V. R., Huffman, J. A., Burrows, S. M., Hoose, C., Safatov, A. S., Buryak, G. A.,
Fröhlich-Nowoisky, J., Elbert, W., Andreae, M. O., Pöschl, U., and Jaenicke, R.: Primary
Biological Aerosol Particles in the Atmosphere: A Review, Tellus Series B-Chemical and
Physical Meteorology, 64, 15598, 2012.
Douwes, J., Thorne, P., Pearce, N., and Heederik, D.: Bioaerosol health effects and exposure
assessment: Progress and prospects, Annals of Occupational Hygiene, 47, 187-200, 2003.
Foot, V. E., Kaye, P. H., Stanley, W. R., Barrington, S. J., Gallagher, M., and Gabey, A.: Low-
cost real-time multi-parameter bio-aerosol sensors, Proceedings of the SPIE - The International
Society for Optical Engineering, 7116, 711601, 2008.
Gabey, A. M., Gallagher, M. W., Whitehead, J., Dorsey, J. R., Kaye, P. H., and Stanley, W. R.:
Measurements and comparison of primary biological aerosol above and below a tropical forest
canopy using a dual channel fluorescence spectrometer, Atmospheric Chemistry and Physics, 10,
460 4453-4466, 2010.

Hairston, P. P., Ho, J., and Quant, F. R.: Design of an instrument for real-time detection of
bioaerosols using simultaneous measurement of particle aerodynamic size and intrinsic
fluorescence, Journal of Aerosol Science, 28, 471-482, 1997.
Hale, G. E. and Wadsworth, F. L. O.: The Modern Spectroscope. XIX. The Objective
Spectroscope, The Astrophysical Journal, 4, 54, 1896.
Healy, D. A., Huffman, J. A., O'Connor, D. J., Pohlker, C., Poschl, U., and Sodeau, J. R.:
Ambient measurements of biological aerosol particles near Killarney, Ireland: a comparison
between real-time fluorescence and microscopy techniques, Atmospheric Chemistry and Physics,
469 14, 8055-8069, 2014.



Hernandez, M., Perring, A., McCabe, K., Kok, G., Granger, G., and Baumgardner, D.:
Composite Catalogues of Optical and Fluorescent Signatures Distinguish Bioaerosol Classes,
Atmos. Meas. Tech. Discuss., 2016, 1-17, 2016.
Hill, S. C., Doughty, D., and Wetmore, A.: Raman Spectra of Individual Particles for
Characterization of Atmsopheric Particles, 34th Annual Conference of the American Association
of Aerosol Research (AAAR), Minneapolis, MN, 2015.
Hill, S. C., Mayo, M. W., and Chang, R. K.: Fluorescence of Bacteria, Pollens, and Naturally
Occurring Airborne Particles: Excitation/Emission Spectra. Laboratory, A. R. (Ed.), Adelphi,
MD, 2009.
Huffman, J. A., Prenni, A. J., DeMott, P. J., Pöhlker, C., Mason, R. H., Robinson, N. H.,
Fröhlich-Nowoisky, J. F., Tobo, Y., Després, V., Garcia, E., Gochis, D. J., Harris, E., Müller-
Germann, I., Ruzene, C., Schmer, B., Sinha, B., Day, D. A., Andreae, M. O., Jimenez, J. L.,
Gallagher, M., Kreidenweis, S. M., Bertram, A. K., and Pöschl, U.: High concentrations of
biological aerosol particles and ice nuclei during and after rain, Atmospheric Chemistry and
Physics, 13, 6151-6164, 2013.
Huffman, J. A., Sinha, B., Garland, R. M., Snee-Pollmann, A., Gunthe, S. S., Artaxo, P., Martin,
S. T., Andreae, M. O., and Poeschl, U.: Size distributions and temporal variations of biological
aerosol particles in the Amazon rainforest characterized by microscopy and real-time UV-APS
fluorescence techniques during AMAZE-08, Atmospheric Chemistry and Physics, 12, 11997-
489 12019, 2012.

Huffman, J. A., Treutlein, B., and Pöschl, U.: Fluorescent biological aerosol particle
concentrations and size distributions measured with an Ultraviolet Aerodynamic Particle Sizer
(UV-APS) in Central Europe, Atmospheric Chemistry and Physics, 10, 3215-3233, 2010.
Jenkins, F. A. and White, H. E.: Fundamentals of optics, Tata McGraw-Hill Education, 1957.
Kaye, P. H., Stanley, W. R., Hirst, E., Foot, E. V., Baxter, K. L., and Barrington, S. J.: Single
particle multichannel bio-aerosol fluorescence sensor, Optics Express, 13, 3583-3593, 2005.
Kiselev, D., Bonacina, L., and Wolf, J.-P.: Individual bioaerosol particle discrimination by multi-
photon excited fluorescence, Optics Express, 19, 24516-24521, 2011.
Lakowicz, J. R.: Principles of Fluorescence Specrtroscopy, Springer, 2010.
Manninen, A., Putkiranta, M., Rostedt, A., Saarela, J., Laurila, T., Marjamaki, M., Keskinen, J.,
and Hernberg, R.: Instrumentation for measuring fluorescence cross sections from airborne
microsized particles, Applied Optics, 47, 110-115, 2008.
Maxwell, K. and Johnson, G. N.: Chlorophyll fluorescence—a practical guide, Journal of
Experimental Botany, 51, 659-668, 2000.
Möhler, O., DeMott, P. J., Vali, G., and Levin, Z.: Microbiology and atmospheric processes: the
role of biological particles in cloud physics, Biogeosciences, 4, 1059-1071, 2007.



Morris, C. E., Conen, F., Huffman, J. A., Phillips, V., Pöschl, U., and Sands, D. C.:
Bioprecipitation: a feedback cycle linking Earth history, ecosystem dynamics and land use
through biological ice nucleators in the atmosphere, Global Change Biology, 20, 341-351, 2014.
O'Connor, D. J., Iacopino, D., Healy, D. A., O'Sullivan, D., and Sodeau, J. R.: The intrinsic
fluorescence spectra of selected pollen and fungal spores, Atmospheric Environment, 45, 6451-
511    6458, 2011.

O'Connor, D. J., Lovera, P., Iacopino, D., O'Riordan, A., Healy, D. A., and Sodeau, J. R.: Using
spectral analysis and fluorescence lifetimes to discriminate between grass and tree pollen for
aerobiological applications, Analytical Methods, 6, 1633-1639, 2014.
Pan, Y.-L., Hill, S. C., Pinnick, R. G., House, J. M., Flagan, R. C., and Chang, R. K.: Dual-
excitation-wavelength fluorescence spectra and elastic scattering for differentiation of single
airborne pollen and fungal particles, Atmospheric Environment, 45, 1555-1563, 2011.
Pan, Y.-L., Huang, H., and Chang, R. K.: Clustered and integrated fluorescence spectra from
single atmospheric aerosol particles excited by a 263-and 351-nm laser at New Haven, CT, and
Adelphi, MD, Journal of Quantitative Spectroscopy & Radiative Transfer, 113, 2213-2221,
521    2012.

Pan, Y. L., Hartings, J., Pinnick, R. G., Hill, S. C., Halverson, J., and Chang, R. K.: Single-
particle fluorescence spectrometer for ambient aerosols, Aerosol Sci. Technol., 37, 628-639,
524    2003.

Pan, Y. L., Hill, S. C., Pinnick, R. G., Huang, H., Bottiger, J. R., and Chang, R. K.: Fluorescence
spectra of atmospheric aerosol particles measured using one or two excitation wavelengths:
Comparison of classification schemes employing different emission and scattering results, Optics
Express, 18, 12436-12457, 2010.
Perring, A. E., Schwarz, J. P., Baumgardner, D., Hernandez, M. T., Spracklen, D. V., Heald, C.
L., Gao, R. S., Kok, G., McMeeking, G. R., McQuaid, J. B., and Fahey, D. W.: Airborne
observations of regional variation in fluorescent aerosol across the United States, J. Geophys.
Res.-Atmos., 120, 1153-1170, 2015.
Pinnick, R. G., Fernandez, E., Rosen, J. M., Hill, S. C., Wang, Y., and Pan, Y. L.: Fluorescence
spectra and elastic scattering characteristics of atmospheric aerosol in Las Cruces, New Mexico,
USA: Variability of concentrations and possible constituents and sources of particles in various
spectral clusters, Atmospheric Environment, 65, 195-204, 2013.
Pöhlker, C., Huffman, J. A., Förster, J.-D., and Pöschl, U.: Autofluorescence of atmospheric
bioaerosols: spectral fingerprints and taxonomic trends of pollen, Atmospheric Measurement
Techniques, 13, 3369-3392, 2013.
Pöhlker, C., Huffman, J. A., and Pöschl, U.: Autofluorescence of atmospheric bioaerosols -
fluorescent biomolecules and potential interferences, Atmospheric Measurement Techniques, 5,
542    37-71, 2012.



Pöschl, U., Martin, S. T., Sinha, B., Chen, Q., Gunthe, S. S., Huffman, J. A., Borrmann, S.,
Farmer, D. K., Garland, R. M., Helas, G., Jimeney, J. L., King, S. M., Manzi, A., Mikhailov, E.,
Pauliquevis, T., Petters, M. D., Prenni, A. J., Roldin, P., Rose, D., Schneider, J., Su, H., Zorn, S.
R., Artaxo, P., and Andreae, M. O.: Rainforest Aerosols as Biogenic Nuclei of Clouds and
Precipitation in the Amazon, Science, 329, 1513-1516, 2010.
Pöschl, U. and Shiraiwa, M.: Multiphase Chemistry at the Atmosphere–Biosphere Interface
Influencing Climate and Public Health in the Anthropocene, Chemical Reviews, 115, 4440-4475,
550 2015.

Rasband, W.: ImageJ, US National Institutes of Health, Bethesda, Maryland, USA, h ttp, imagej.
nih. gov/ij, 2012, 1997.
Robinson, N. H., Allan, J. D., Huffman, J. A., Kaye, P. H., Foot, V. E., and Gallagher, M.:
Cluster analysis of WIBS single-particle bioaerosol data, Atmospheric Measurement Techniques,
555 6, 337-347, 2013.

Ronningen, T., Schuetter, J., Wightman, J., and Murdock, A.: Raman spectroscopy for biological
identification, Biological Identification: DNA Amplification and Sequencing, Optical Sensing,
Lab-On-Chip and Portable Systems, 2014. 313, 2014.
Saari, S., Reponen, T., and Keskinen, J.: Performance of Two Fluorescence-Based Real-Time
Bioaerosol Detectors: BioScout vs. UVAPS, Aerosol Science and Technology, 48, 371-378,
561 2014.

Saari, S., Reponen, T., and Keskinen, J.: Performance of Two Fluorescence-Based Real-Time
Bioaerosol Detectors: BioScout vs. UVAPS, Aerosol Sci. Technol., 48, 371-378, 2013.
Schumacher, C. J., Pöhlker, C., Aalto, P., Hiltunen, V., Petäjä, T., Kulmala, M., Pöschl, U., and
Huffman, J. A.: Seasonal cycles of fluorescent biological aerosol particles in boreal and semi-
arid forests of Finland and Colorado, Atmos. Chem. Phys., 13, 11987-12001, 2013.
Sivaprakasam, V., Huston, A. L., Scotto, C., and Eversole, J. D.: Multiple UV wavelength
excitation and fluorescence of bioaerosols, Optics Express, 12, 4457-4466, 2004.
Sivaprakasam, V., Pletcher, T., Tucker, J. E., Huston, A. L., McGinn, J., Keller, D., and
Eversole, J. D.: Classification and selective collection of individual aerosol particles using laser-
induced fluorescence, Applied Optics, 48, B126-B136, 2009.
Xiong, B., Zhou, R., Hao, J., Jia, Y., He, Y., and Yeung, E. S.: Highly sensitive sulphide
mapping in live cells by kinetic spectral analysis of single Au-Ag core-shell nanoparticles, Nat
Commun, 4, 1708, 2013.




**Figures**

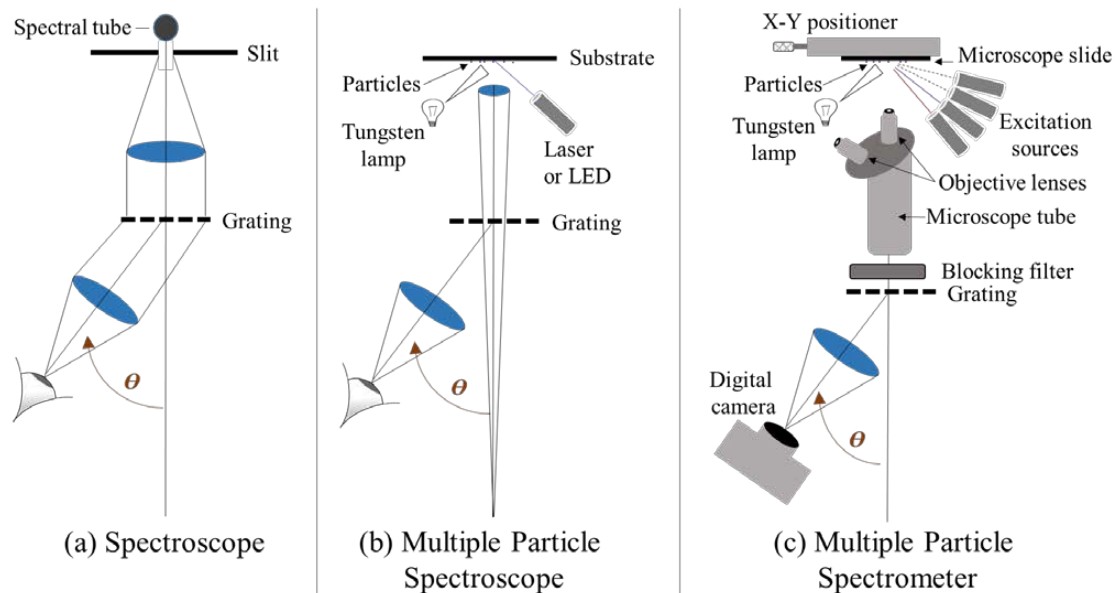

(a) Spectroscope    (b) Multiple Particle Spectroscope    (c) Multiple Particle Spectrometer

Figure 1: Three-stage progression of spectrometer. (a) Spectroscope, as often utilized in student
laboratories. (b) Multiple particle spectroscope. (c) Multiple particle spectrometer (introduced
here).

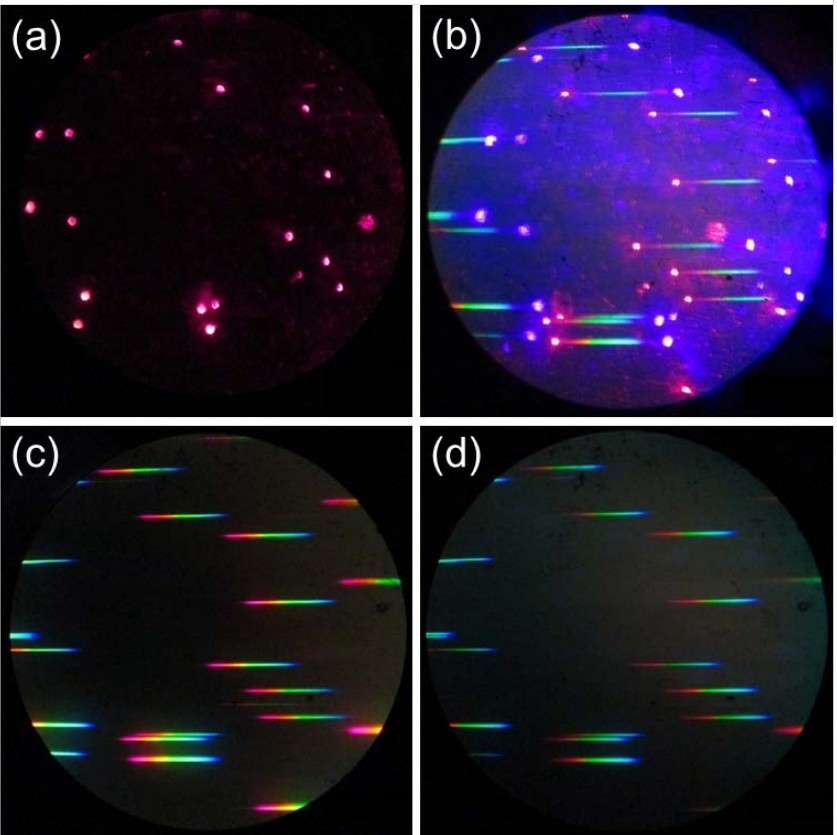

Figure 2: Four-panel progression of images acquired for a given scene of paper mulberry pollen
particles collected onto a glass microscope slide. Scale is the same in each figure, with each
horizontal swath of color approximately 10 µm in height. (a) Dark field image of particles
illuminated by monochromatic red laser light ($\Theta = 0$). (b) Particles illuminated with both violet
(405 nm) and red (650 nm) diode lasers. Fluorescent spectra of individual particles showing
image taken without use of blocking filter. (c) White light illumination with tungsten filament
bulb. (d) Fluorescent emission with excitation from violet diode laser, but using blocking filter to
remove violet laser point.



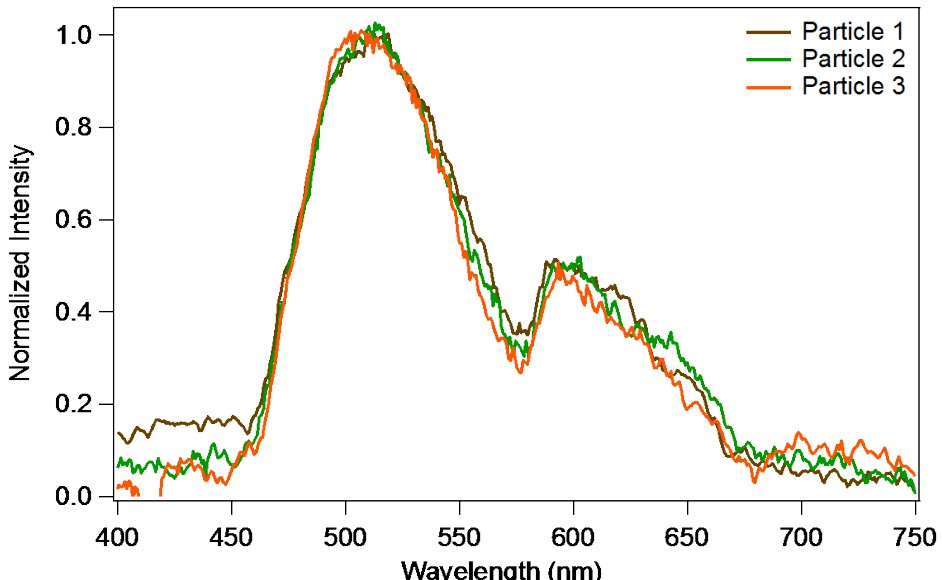

Figure 3: Fluorescent spectra of three individual paper mulberry pollen particles (*Broussonetia*
*payrifera*) illuminated by 405 nm diode laser (Fig. 2d). Emission wavelength was calibrated
using 405 nm and 650 nm laser points (Fig. 2b). All spectra were normalized to 1.0 maximum
peak height.





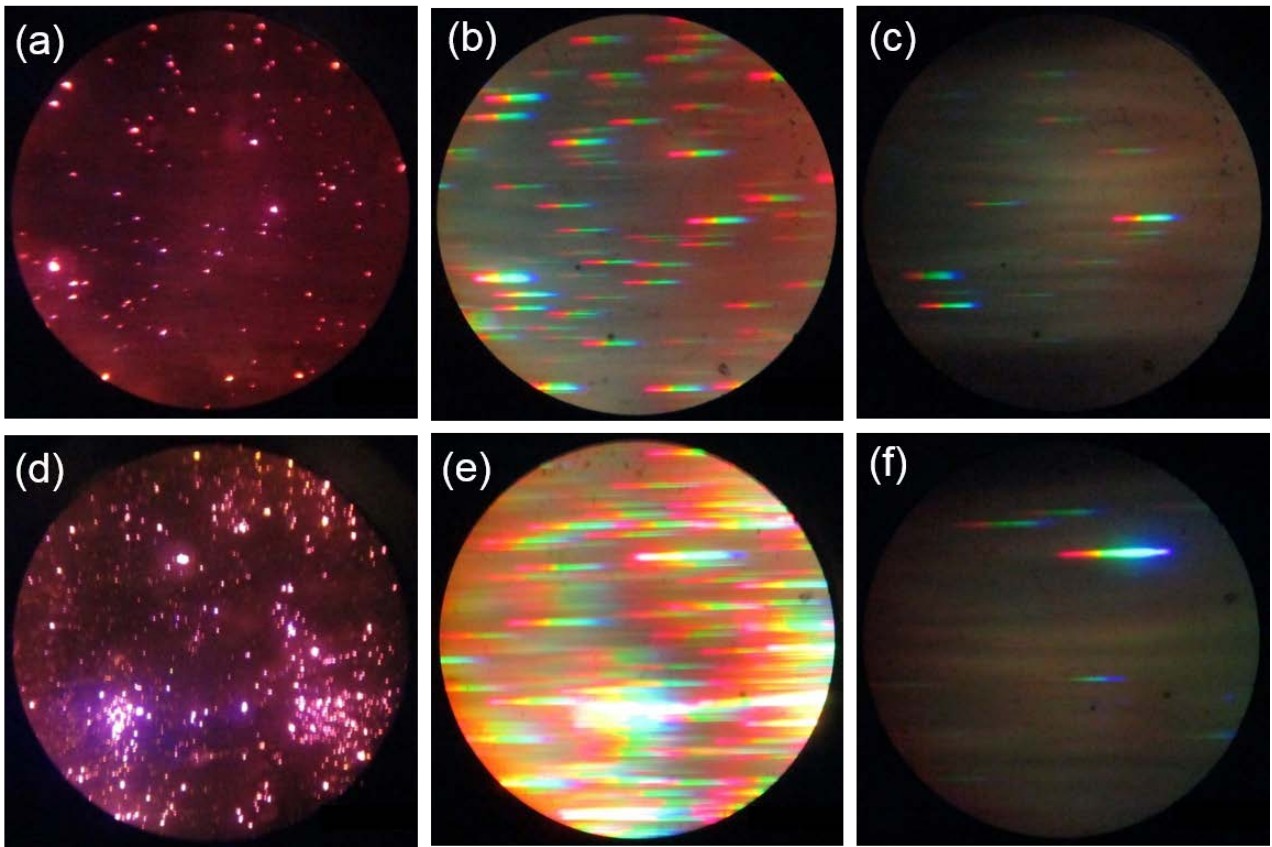

Figure 4: Images showing minority of fluorescent particles amidst large quantity of other
particles. (a-c; top panels) outdoor ambient particles collected via natural settling onto
microscope slide. (d-f; bottom panels) ground optical fused quartz particles. First column (a,d)
shows conventional micrograph images ($\Theta = 0$) illuminated with red laser light. Second column
(b,e) shows white light scattering spectra after illumination of same scene with polychromatic
light from tungsten source. Third column (c,f) shows fluorescent spectra after illumination of
same scene with 405 nm diode laser and blocking filter in place.




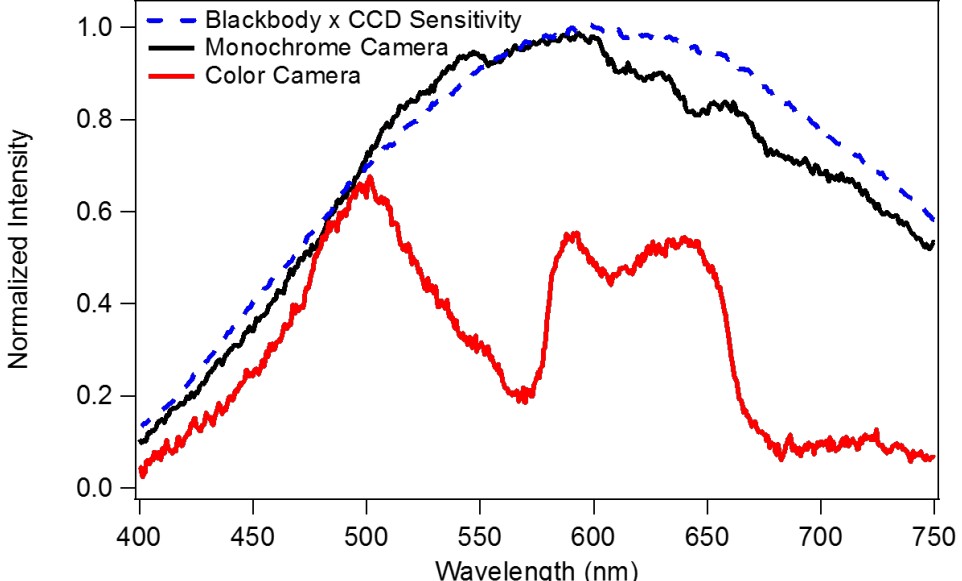

Figure 5: Comparison of white light scattering from ground sodium chloride (NaCl) particle
detected by color (solid red) and monochrome (solid black) cameras. Reference spectrum
(dashed blue) shows calculated blackbody radiator at 3000 K multiplied by CCD sensitivity
curve as an estimate of light intensity as a function of wavelength expected at detector. Red
curve was normalized to 1.0 and then scaled down by an arbitrary value to show reduced
emission intensity above 500 nm.



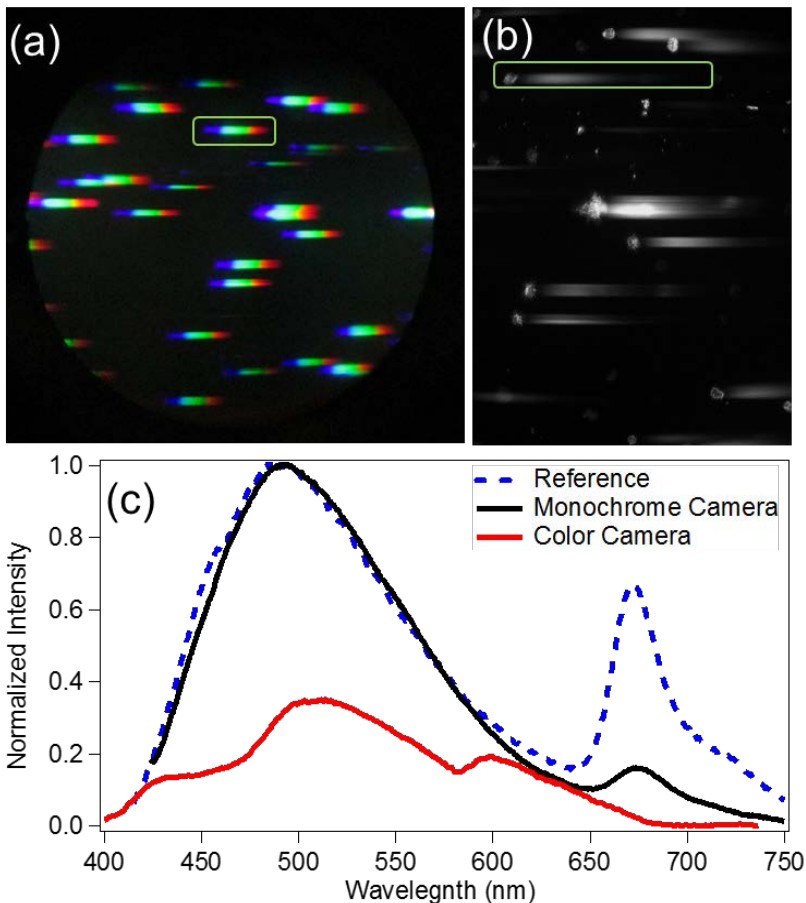

Figure 6: Spectral images and associated spectra for individual particles of Kentucky bluegrass
(*Poa pratensis*) pollen. Both images show fluorescence of particles illuminated by 405 nm diode
laser and utilizing 420 nm long-pass blocking filter. (a) Color camera detection. (b) Black-and-
white camera detection. (c) Spectra of one particle from each image (boxed particle from a,b
shown). Reference spectrum (dashed line) from microplate reader spectrofluorometer showing
bulk average of ~5 mg of material. Spectra from monochrome camera and reference technique
normalized to peak height of 1.0. Spectrum from color camera arbitrarily scaled to show reduced
intensity of collection compared to monochrome camera.

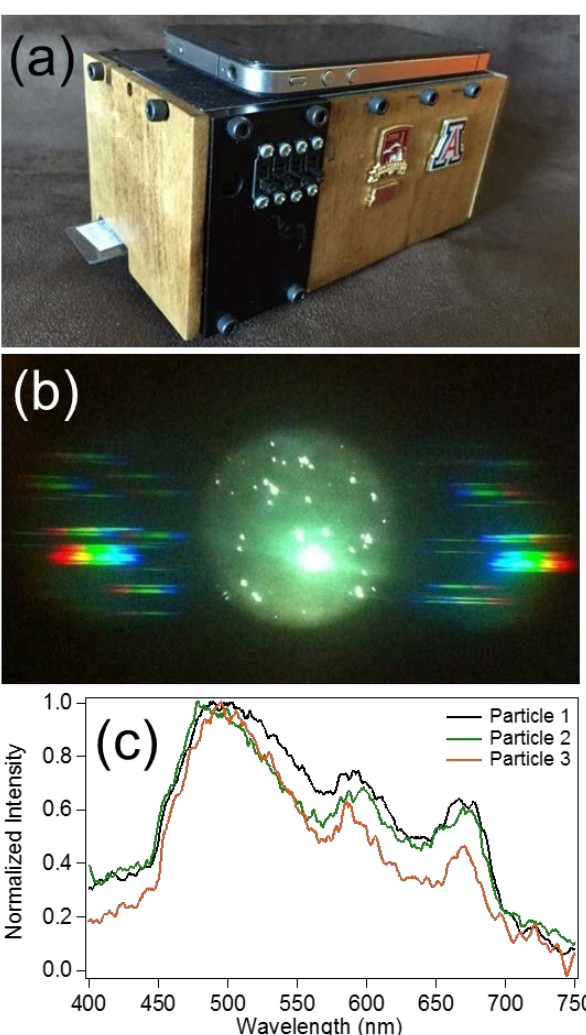

Figure 7: Smartphone spectrofluorometer prototype. (a) Photograph of iPhone spectrometer. (b)
Standard micrograph (dots) and fluorescent swaths (left and right portions) collected as single
image with iPhone 5s shown in panel (a). Particles are Kentucky bluegrass pollen (*Poa*
*pratensis*) (c) Spectra of three of the smaller particles from panel (b), normalized to peak height
of 1.0.