# Peer review of "Atmos. Meas. Tech. Discuss., doi:10.5194/amt-2016-153, 2016 Manuscript under review for journal Atmos. Meas. Tech. Published: 31 May 2016"

_Atmospheric Measurement Techniques, 2016_

## Referee Comment (RC1) · Anonymous Referee #1 · 25 Jun 2016

General Comments This manuscript is interesting, important and well written. I like it. It appears to be a major step forward in developing low cost instrumentation for aerosols, especially biological aerosols. Because of the low cost I suspect that, as the authors suggest, versions of this instrument will be used to study aerosols over a much larger spatial range than possible with presently available instruments. Present instruments are too expensive. The potential for making apps for cellphones to record the spectra and send these to one location for assembling the data from all the sensors is appealing. This is first I remember seeing the suggestion to spectrally disperse the emission from aerosol particles spread randomly in 2D. I recommend publication and do not suggest any mandatory changes.

[Figure]

Specific Comments The authors may want to think about, and possibly comment on, the following. Possibly more could be said about the smaller end of the size range of biological particles that could be detected. What is the large dimension of the smallest particles measured? Could a 1 micron bit of a fungal spore be detected? As compared to illuminating with a line source that must be stepped in one direction over the image, this approach needs no moving parts. What is given up for this advantage? Is the maximum number of particles per area that could be analyzed lower? I think yes. Is the spectral range less? Again, I think yes. That probably isn't so important for fluo-rescence because the bands are not sharp so 20 wavelength bands may be adequate. Raman was mentioned. In Raman spectroscopy the light from 0 to 4000 cm-1 might be spread over 1000 pixels or so when illuminating with a line source. That requires sig-nificant distance on the camera. I wonder if the problem of overlapping spectra would make this multi-particle spectrometer approach unworkable for Raman in cases where a large wavenumber range is desired.

Technical Corrections 586, 592, 602 "fluorescent spectra", should be changed to "fluo-rescence spectra" as in every other time it occurs in the paper. 425 "grass-type pollens (i.e. Ambrosia or ragweed)"? Ambrosia is not a grass. It is in Compositae (Aster family). If ragweed is in a grass-type pollen group, I suggest a citation for "grass-type pollen."
* * *

---

## Referee Comment (RC2) · Anonymous Referee #2 · 6 Jul 2016

This manuscript describes the development of a new instrument to obtain scattering and fluorescence spectra from individual aerosol particles collected on a microscope slide. The new technology will certainly be of interest to the atmospheric science community and the manuscript is generally well structured however I think it could say more about certain aspects of the technology and the implementation. Therefore I recommend publication after the following comments have been addressed:

1. It seems that the size range of particles detectable by this instrument is a critical piece of information that is currently not addressed quantitatively. The authors state that they are targeting "micron-sized" particles, however, all of the known particles that they look at are pollen species which are significantly larger than a micron.

[Figure]

Can the authors show what it looks like when this technology is applied to samples of smaller particles such as bacteria, spores or man-made size-selected particles such as polystyrene latex spheres? Since detection of spores seems to be one of the main motivations it would be nice to show that this instrument can work with something other than pollen. Along similar lines, I believe the authors state that the height of the swath is related to the particle size. More explicit discussion of this relationship would be helpful.

2. In general, it would be nice if all of the graphics could be accompanied with a quantitative statement of what is "found" in the view graph. For example, in figure 2, what percentage of the particles appearing in panel a result in a spectrum in panel d? Clearly it is most of them but it would be nice to know if it's 100% or something less than that. Then in figure 4, quantitative information is given for the top panels but not for the bottom. Here it would be nice to know how many quartz particles are identified in the viewgraph and what fraction of that number the "fluorescent needle in the haystack" contributes. If only 10% of all particles are identified as fluorescent in an ambient sample, then a "false positive" rate of even a few percent could be significant.

3. Related to comment 1 above, the functional minimum size for fluorescent detection may also not be a limitation purely of how small a particle can be imaged through the microscope optics but, rather, how much fluorophore a particle must contain to yield a detectable spectrum given the hardware. What is the primary limitation to detection of "less bright" fluorescent things? For example, in Figure 4c, I can see the 7 spectra discussed in the paper but I can also see 5 or 6 other, more faint spectra that could also be fluorescent particles. How have the authors determined the intensity threshold required to call a particle fluorescent?

4. Can the authors provide the dimensions (distance to camera and angle theta for collection) of their two instrumental set-ups along with the imaging area and pixel size of the cameras and phones used? I believe the combination of these choices is what determines the spectral resolution achieved and it would be nice to walk the reader

through these relationships.

5. In section 4, I don't follow why a 3000 k blackbody spectrum is used to approximate a theoretical scattering curve for NaCl. Is that supposed to read 300 k? If so the same type-o occurs in the legend of Figure 5.

———————————————

---

## Author Comment (AC1) · 29 Jul 2016

**Response to referee comment on amt-2016-153 by Huffman et al.**

**Anonymous Referee #1**

General Comments: This manuscript is interesting, important and well written. I like it. It appears to be a major step forward in developing low cost instrumentation for aerosols, especially biological aerosols.

Because of the low cost I suspect that, as the authors suggest, versions of this instrument will be used to study aerosols over a much larger spatial range than possible with presently available instruments.

Present instruments are too expensive. The potential for making apps for cellphones to record the spectra and send these to one location for assembling the data from all the sensors is appealing. This is first I remember seeing the suggestion to spectrally disperse the emission from aerosol particles spread randomly in 2D. I recommend publication and do not suggest any mandatory changes.

Author response: We thank the referee for his/her positive assessment and summary. We have indeed not seen an instrument that offers the range of capabilities as the one introduced in our manuscript and we are excited to further the development of the technology.

Note regarding document formatting: black text shows original referee comment, blue text shows author response, and red text shows quoted manuscript text. Changes to manuscript text are shown as *highlighted and underlined*. All line numbers refer to discussion/review manuscript.

Specific Comments (note that referee comments have been labeled by letter and chopped by individual referee-thought so they can be dealt with in a clear sequence): The authors may want to think about, and possibly comment on, the following. **[a]** Possibly more could be said about the smaller end of the size range of biological particles that could be detected.

The referee brings up some really good points here. Even though we highlight the positive attributes of the technique we introduced, there are always disadvantages and trade-offs to consider. The points the referee mentions are some of these. Based on the tone and text of the referee comment, we would guess that s/he would agree that a deep analysis of these trade- offs is beyond the scope of this manuscript, but we decided to add a few additional overview statements to the manuscript to make it clear that we acknowledge these important trade-offs.

In particular, we added Section 2.3 (before L228) that discusses some practical considerations brought up by the referee and we also added a paragraph to the conclusions (L427)

summarizing the novel benefits of the technique. These two additional paragraphs are copied in this document at line 181.

Other responses to specific points raised by the referee:

**[a]** First, the originally submitted manuscript referred to the device investigating "micron-size particles." These statements have been changed to "*approximately super*micron-size particles"

in L13 and L71 as also discussed in response to Referee #2 (Point **[1a]**).

Second, a rigorous discussion of the lower size limit of detectable particles is complex, because it convolves several instrument parameters. A deeper discussion of this is presented in response to Referee #2 (Point **[1a,b,c]**). In short, however, we have investigated particles as small as ~1

μm, and we are confident that the technique will also work for particles smaller than this. The lower limit will depend strongly on the relative fluorescence intensity of the particle and the exposure time of the camera, among several other factors. We have not yet rigorously probed the interplay of these variables, but will continue to do so as experimental development work continues. In response to the comments from both referees, however, we added supplemental Figure S2 and associated text at L253 discussing micrographs and spectra associated with 1 μm fluorescent polystyrene latex beads interrogated by our benchtop device:

> "This fraction is highly dependent on the threshold one applies to categorize a given particle as fluorescent or not. *Observed* fluorescence intensity is *also* strongly a function of *several factors, including*: particle size, *fluorophore content and quantum yield, intensity of excitation source, instrument optics, and camera exposure time (e.g. Hill et al., 2001; Hill et al., 2013; Hill et al., 2015b; Pöhlker et al., 2012; Sivaprakasam et al., 2011)*. Most fluorescence-based aerosol detectors are faced with the *conceptual* challenge of *how best to define minimum detectable fluorescence*, and the sensitivity of a given detector will significantly influence the comparison of the relative fraction of fluorescent particles detected by any two instruments or types of instruments (*e.g.* Healy et al., 2014; Hernandez et al., 2016; Huffman et al., 2012; Saari et al., 2013). *As mentioned, the particle size contributes significantly to the detectability of fluorescence from individual particles. All particles chosen for discussion here are relatively large (*e.g. >10 μm*) in order to highlight the overall technique and concepts. It should be noted, however, that the instrument is not fundamentally limited to such large particles and can be applied to particles of 1 μm in diameter, or smaller, if higher microscope magnification (e.g. 40x) is utilized and the parameters influencing observed fluorescence are managed appropriately. We have acquired spectra of individual particles as small as 0.96 μm (e.g. supplemental Fig. S2), though this is not intended to be presented as a lower limit. Further limitations will be explored in follow-up studies.*"

References added here:

Hill, S. C., Williamson, C. C., Doughty, D. C., Pan, Y. L., Santarpia, J. L., and Hill, H. H.: Size-dependent fluorescence of bioaerosols: Mathematical model using fluorescing and absorbing molecules in bacteria, Journal of Quantitative Spectroscopy & Radiative Transfer, 157, 54-70, 2015b.

Hill, S. C., Pan, Y. L., Williamson, C., Santarpia, J. L., and Hill, H. H.: Fluorescence of bioaerosols: mathematical model including primary fluorescing and absorbing molecules in bacteria, Optics Express, 21, 22285-22313, 2013.

Hill, S. C., Pinnick, R. G., Niles, S., Fell, N. F., Pan, Y. L., Bottiger, J., Bronk, B. V., Holler, S., and Chang, R. K.: Fluorescence from airborne microparticles: dependence on size, concentration of fluorophores, and illumination intensity, Applied Optics, 40, 3005-3013, 2001.

Sivaprakasam, V., Lin, H.-B., Huston, A. L., and Eversole, J. D.: Spectral characterization of biological aerosol particles using two-wavelength excited laser-induced fluorescence and elastic scattering measurements, Optics Express, 19, 6191-6208, 2011.

**[*b*]** What is the large dimension of the smallest particles measured?

**[*b*]** We are not quite sure what this question is asking, but provide here response that we think
addresses the question. Using Figure 4 as an example, the vertical extent of the elastic (e.g Fig.
4b) and inelastic/fluorescence (e.g. Fig. 4c) spectra shown vary as a function of particle size. For
example, if a particle is large in the vertical (*y*) dimension, the height of its spectral swath will be
approximately equal to the height of the particle itself.

We added the following text at L108 of the manuscript:
"*For example, if a particle is large in the vertical (y) dimension, the height of its spectral*
*swath will be approximately equal to the vertical dimensions of the particle itself.*"

**[*c*]** Could a 1 micron bit of a fungal spore be detected?

**[*c*]** Yes, a 1 μm fungal spore could be detected, as long as it is "sufficiently" fluorescent and the
exposure time of the camera is set appropriately. See response to Point **[*a*]**.

**[*d*]** As compared to illuminating with a line source that must be stepped in one direction over the image,
this approach needs no moving parts. What is given up for this advantage?

**[*d*]** One technical disadvantage of the method described here is that spectral resolution in the
'x-direction' (i.e. the dimension into which the spectrum is dispersed) is reduced when analyzing
a large particle. The reason for this is as follows. Assume an illumination source is a line of
infinitesimal width, shining across the whole field of view in the y-direction (i.e. top-to-bottom
on Fig. 2), and scanning slowly from left to right. As it scans, the source will hit the left side of a
given particle and disperse fluorescence emitted from that small portion of material (dx) into
the x-direction. As the scan line moves to the right it will excite a fluorescence spectrum from a
different small piece (dx) of material. The angle of dispersion (*Θ*) for a given wavelength (color)
of light emitted is a constant, however. Thus, fluorescence emitted from the first point at one
emission wavelength will be convolved into the emission spectrum from a second physical point
of excitation, but at a different emission wavelength. This will blur the fluorescence spectrum in
wavelength space increasingly as a function of particle size. Additionally, if a given particle is
homogeneous in composition, the fluorescence spectrum will not vary as the illuminating line
traverses the width of the particle. If a particle is inhomogeneous, however, the fluorescence
spectrum may change as the illumination point moves, further smearing the fluorescence
spectrum. Fortunately, as the referee points out, the emission bands for fluorescence spectra
are broad, and the extent of this smearing is small for particles e.g. < 50 μm.

**[*e*]** Is the maximum number of particles per area that could be analyzed lower? I think yes.

**[*e*]** The short answer here is yes, the maximum number of particles analyzed by the technique as
presented is theoretically lower than a hypothetical technique that utilizes stepped-line
illumination. This is because, when all particles in a field of view are illuminated at the same
time, the emission spectrum from one particle may be projected onto a location that overlaps
with another particle. Illuminating particles individually would reduce this issue. The point of
this concept, however, is to create a simple and inexpensive device to produce information
about fluorescence of individual particles. As the referee points out, adding a stepping illumination line would introduce either moving parts or more complicated components and would also increase analysis complexity.

**[*f*]** Is the spectral range less? Again, I think yes.

> **[*f*]** Again, the short answer is yes, the spectra range of the device discussed here is theoretically reduced by illuminating all particles at once. This is because the emission spectrum of one particle can be projected in the x-dimension such that it can overlap with the emission spectrum of another particle. The wider the spectral range of interest, the further individual particles must be separated to be able to illuminate them simultaneously.

**[*g*]** That probably isn't so important for fluorescence because the bands are not sharp so 20 wavelength bands may be adequate. Raman was mentioned. In Raman spectroscopy the light from 0 to 4000 cm-1 might be spread over 1000 pixels or so when illuminating with a line source. That requires significant distance on the camera. I wonder if the problem of overlapping spectra would make this multi-particle spectrometer approach unworkable for Raman in cases where a large wavenumber range is desired.

> **[*g*]** Again, yes. As mentioned above, the fact that fluorescence bands are naturally broad reduces the requirements for high resolution. In concept, the device could be applied to the acquisition and analysis of Raman spectra, though there are a whole host of practical challenges associated with this extension of the idea. One of these challenges is that Raman spectra are fundamentally much narrower than fluorescence spectra, and thus, to acquire a Raman spectra with any reasonable level of resolution would require much higher resolution than would be required to achieve the fluorescence spectra discussed here. So it is possible that this technique could not practically be applied to Raman spectra. We very briefly introduced the idea as a tantalizing future possibility, but tried to do so in a way that did not promise that it would work. Based on the referee's valid comment we amended the statements in the manuscript (mostly in the final paragraph of Section 6: A vision for broad scale use) as follows:
>
> > (Starting L402): "The technique of acquiring spectra from individual particles _could_ perhaps also be applied to the acquisition of Raman scattering spectra, though this _would_ introduce additional technical challenges _such as the need for relatively high spectral resolution, which is compromised in our slitless spectrometer technique_. Recently an instrument for real-time detection of single particles in air by Raman spectroscopy has been made commercially available (Hill et al., 2015_a_; Ronningen et al., 2014). The instrument described here could be developed in the future to provide Raman spectroscopy of individual atmospheric particles, with reduced resolution or signal-to-noise, but also with significantly reduced cost. _The development of a Raman-oriented instrument would require significant future development, however._"
>
> Text added to before L228 as new Section 2.3:
> > _"As a practical matter, the density of particles distributed on the slide should be sufficiently sparse that the spectral swaths do not overlap if individual particle spectra are to be determined. This requirement arises as a result of the fact that the entire field of view is illuminated at once, ideally exciting many, e.g. 5-30, particles. The wider the spectral range desired, the more this effect is enhanced. This particle density limitation is diminished, however, if one is only interested primarily in the relative fraction of particles that fluoresce at a given excitation wavelength._

*The technique introduced here also presents fundamental limitations in spectral resolution*
*influenced, in part, by particle size and homogeneity. For example, fluorescence emitted*
*from the near side of a large particle at a given wavelength and Ө angle will be dispersed at*
*the same Ө angle to a dissimilar point in the color swath from the far side of the same*
*particle. This will blur the fluorescence spectrum in wavelength space, increasingly as a*
*function of particle size. Additionally, if a given particle is inhomogeneous in composition,*
*the fluorescence spectrum emitted by two points on the particle will be dissimilar, and thus*
*the resultant spectrum will be smeared somewhat. Fluorescence emission bands are*
*fundamentally broad and smooth, however, and so the extent of the associated smearing*
*due to particle size or inhomogeneity does not practically impact the observed spectra for*
*particles that are smaller than many tens of microns."*

Text added to conclusions (at L428) summarizing advantages and disadvantages of the
technique and to address many of the comments introduced by the referee:
"*The strong benefits of the described technique include that many particles can be analyzed*
*simultaneously and that fluorescence spectra can be rapidly acquired for individual*
*particles, each at multiple wavelengths, and at a cost potentially orders of magnitude lower*
*than existing techniques. Further, the technique provides the possibility to probe at a glance*
*for contamination of fluorescent particles that could contaminate a collection of non-*
*fluorescent material, even without needing to analyze spectra."*

Technical Corrections 586, 592, 602 "fluorescent spectra", should be changed to "fluorescence spectra"
as in every other time it occurs in the paper.

All changed.

425 "grass-type pollens (i.e. Ambrosia or ragweed)"? Ambrosia is not a grass. It is in Compositae (Aster
family). If ragweed is in a grass-type pollen group, I suggest a citation for "grass-type pollen."

This is a good catch by the referee and a mistake on our part.  We changed the statement in this
case to say "grass-type pollens (i.e. *Dactylis glomerata* or *Orchard grass*) …".  As re-written the
existing citations are sufficient.

---

## Author Comment (AC2) · 29 Jul 2016

**Response to referee comment on amt-2016-153 by Huffman et al.**
**Anonymous Referee #2**
This manuscript describes the development of a new instrument to obtain scattering and fluorescence
spectra from individual aerosol particles collected on a microscope slide. The new technology will
certainly be of interest to the atmospheric science community and the manuscript is generally well
structured however I think it could say more about certain aspects of the technology and the
implementation. Therefore I recommend publication after the following comments have been
addressed:
Author response: We thank the referee for his/her positive assessment and recommendation
for publication. The comments that s/he has brought up below are good ones, and the changes
we have processed as a result have improved the quality and clarity of the manuscript. In
preparation of the originally submitted manuscript, our goal had been to provide an
introduction to the concept of the aerosol analysis technique, attempting to balance brevity and
sufficient detail. In some case the balance may have been overly concise, lacking a few details
that could help understand some of the process and trade-offs. We have added text in
association with each of the points discussed below.
Note regarding document formatting: black text shows original referee comment, which have
been chopped into individual thoughts, blue text shows author response, and red text shows
quoted manuscript text. Changes to manuscript text are shown as *highlighted and underlined*.
All line numbers refer to discussion/review manuscript.
1. **[1a]** It seems that the size range of particles detectable by this instrument is a critical piece of
information that is currently not addressed quantitatively. The authors state that they are targeting
"micron-sized" particles, however, all of the known particles that they look at are pollen species which
are significantly larger than a micron.
**[1a]** It is true that the size range of particles detectable by this technique is important to
introduce. First, the referee points out that we introduce the technique as detecting "micron-
size particles," but only show particles > 10 µm in diameter. To be somewhat clearer we
changed the wording in both instances it appears to refer to "supermicron-size particles" as
described here:
  L13 (in Abstract): "We describe a novel, low-cost instrument to acquire both elastic and
  inelastic (fluorescent) scattering spectra from individual *super*micron-size particles …"
  L71: "…where a minority of *approximately super*micron-sized particles …"
To a deeper level, a discussion of the lower size limit of detectable particles brings in a much
more complex discussion that we tried to avoid for this initial overview of the detection
technique. The answer here is also inextricably linked to the question posed below, **[1b]** and
**[3a]**.
**[1b]** Can the authors show what it looks like when this technology is applied to samples of smaller
particles such as bacteria, spores or man-made size-selected particles such as polystyrene latex spheres?

Since detection of spores seems to be one of the main motivations it would be nice to show that this
instrument can work with something other than pollen.
[1b] The main point of the manuscript was to introduce the general concept and to provide a
proof-of-concept, but not to explore all physical relationships. As a result we had originally
chosen to show images and spectra representative of certain pollen. The request to show the
response of the instrument to smaller particles is certainly a reasonable request, however, and
we agree that adding this information would improve the manuscript. In response to the
referee's comment we added Supplemental Figure S2 showing images and spectra of 0.96 µm
polystyrene latex beads and we added the following text at L253:
"This fraction is also highly dependent on the threshold one applies to categorize a given
particle as fluorescent or not. *Observed* fluorescence intensity is *also* strongly a function of
*several factors, including*: particle size, *fluorophore content and quantum yield, intensity of*
*excitation source, instrument optics, and camera exposure time (e.g. Hill et al., 2001; Hill et*
*al., 2013; Hill et al., 2015b; Pöhlker et al., 2012; Sivaprakasam et al., 2011)*. Most
fluorescence-based aerosol detectors are faced with the *conceptual* challenge of *how best*
*to define minimum detectable fluorescence*, and the sensitivity of a given detector will
significantly influence the comparison of the relative fraction of fluorescent particles
detected by any two instruments or types of instruments (*e.g.* Healy et al., 2014;
Hernandez et al., 2016; Huffman et al., 2012; Saari et al., 2013). *As mentioned, the particle*
*size contributes significantly to the detectability of fluorescence from individual particles. All*
*particles chosen for discussion here are relatively large (*e.g. >10 µm*) in order to highlight*
*the overall technique and concepts. It should be noted, however, that the instrument is not*
*fundamentally limited to such large particles and can be applied to particles of 1 µm in*
*diameter, or smaller, if higher microscope magnification (e.g. 40x) is utilized and the*
*parameters influencing observed fluorescence are managed appropriately. We have*
*acquired spectra of individual particles as small as 0.96 µm (e.g. supplemental Fig. S2),*
*though this is not intended to be presented as a lower limit. Further limitations will be*
*explored in follow-up studies.*"
[1c] Along similar lines, I believe the authors state that the height of the swath is related to the particle
size. More explicit discussion of this relationship would be helpful.
[1c] We addressed a similar question posed by Referee #1 in Point [b].  The following text was
added at L108:
"*For example, if a particle is large in the vertical (y) dimension, the height of its spectral*
*swath will be approximately equal to the vertical dimensions of the particle itself.*"
2. In general, it would be nice if all of the graphics could be accompanied with a quantitative statement
of what is "found" in the view graph. [2a] For example, in figure 2, what percentage of the particles
appearing in panel a result in a spectrum in panel d? Clearly it is most of them but it would be nice to
know if it's 100% or something less than that.
[2a] The statement requesting quantitative statements about the images is also
understandable. In the case of Figure 2, we previously addressed this idea in the submitted
manuscript by stating the following (L207): "By comparing Figures 2c and 2d one can see that
the relative fraction of pollen particles fluorescent in this sample is nearly 100%, since this particulate sample is made up of a single kind of pollen." To be clearer we changed the text in
the manuscript as follows:
"By comparing Figures 2c and 2d one can see that the relative fraction of pollen particles
fluorescent in this sample is nearly 100%, since this particulate sample is made up of a
single kind of *relatively highly fluorescent particles. A particle-by-particle comparison is*
*somewhat more difficult, because each spectral swath (Figs. 2b-d) extends to the left from*
*the non-dispersed particle location (Fig. 2a). In some cases the swaths of multiple particles*
*overlap, and in other cases the spectrum is dispersed to a point out of the field of view. A*
*more detailed analysis of all individual particles in Figure 2 is described in the online*
*supplemental information (Fig. S1).*"
We also added Figure S1 and associated supplemental text to unambiguously discuss how the
spectrum from each individual particle is presented in the dispersed and undispersed panels of
Figure 2.
**[2b]** Then in figure 4, quantitative information is given for the top panels but not for the bottom. Here it
would be nice to know how many quartz particles are identified in the viewgraph and what fraction of
that number the "fluorescent needle in the haystack" contributes. If only 10% of all particles are
identified as fluorescent in an ambient sample, then a "false positive" rate of even a few percent could
be significant.
**[2b]** First, we added a quantitative assessment to the statement in the text, as copied here:
L266: "Figure 4f, however, shows one unexpected, strongly fluorescent particle and
approximately three *to five* other weakly fluorescent particles *out of the 200-250 particles in the*
*image (e.g. ~2%)*."
More importantly, however, the comment by the referee reveals a misunderstanding that we
realized we need to clarify. It is true that when applied to ambient aerosol sampling, a "false
positive" of a few percent could add significantly to the uncertainty of the overall measurement.
In the case described here using ground quartz, however, the few particles that fluoresce are
introduced as anomalous not because they show erroneous fluorescence as an instrumental
error, but rather the fluorescence exhibited is real and reveals that a small, but non-zero,
fraction of particles exhibit unexpected fluorescence. The point of this example was indeed to
show that one application of the technique may be to probe for fluorescent contaminant
particles in a matrix of predominantly non-fluorescent particles (i.e. "haystack"). So the
quantitative percentage (i.e. ~2% in Fig. 4f) is not the point as much as that individual particles
can be detected despite the fact that there are so many others in the image. Of course, the
quantitative assessment of the fluorescent fraction may be useful, but this will depend on the
threshold at which 'fluorescent particle' is determined, as discussed above.
To clear up some of this we changed the following sentence at L273:
 "*This example* illustrates how *fluorescent* impurities might easily be detected *at a glance*
 with our apparatus even in the presence of a large majority of non-biological, or otherwise
 non-fluorescent particles, *and without needing to go through the extra step to extract the*
 *actual spectra. Analyzing images in this way also removes the restriction of limiting spectral*
 *swaths from overlapping, and enables a user to collect a rather large number of particles in*
 *one field of view (e.g. hundreds) compared to the far smaller number limiting the analysis if*
 *determination of individual spectra is desired. The collection and analysis of particles in this*

*case can be very rapid and yet can provide a powerful diagnostic tool by positively*
*identifying the approximate fraction of fluorescent impurities in a collection of particles."*

3. **[3a]** Related to comment 1 above, the functional minimum size for fluorescent detection may also not
be a limitation purely of how small a particle can be imaged through the microscope optics but, rather,
how much fluorophore a particle must contain to yield a detectable spectrum given the hardware.

**[3a]** In short, the referee is absolutely correct. However, in addition, the concept of detectability
here is actually somewhat more complicated even than the referee points out, because it is
inextricably related to at least five physical parameters: (1) particle size, (2) fluorophore amount
and quantum yield, (3) intensity of excitation source, (4) instrument optics, and (5) detector
exposure time.  This has been dealt with at length in various other publications and was hinted
at in the originally submitted manuscript, however, we have added brief discussion of this
concept more directly, including five associated references.

Based on this, the functional minimum size for fluorescence detection is not a trivial parameter
to rigorously define, but it is at least < 1 μm.  We have added a new figure to the online
supplement that shows images and spectra for 0.96 μm fluorescent polystyrene latex beads
supporting the idea that the lower limit for particle detection is at least smaller than this size.
We feel that a rigorous exploration of these four inter-related variables as it relates to particle
size would be well beyond the scope of the present manuscript. See point **[1b]** for text that was
inserted along with the new supplemental Figure S2.

**[3b]** What is the primary limitation to detection of "less bright" fluorescent things? For example, in
Figure 4c, I can see the 7 spectra discussed in the paper but I can also see 5 or 6 other, more faint
spectra that could also be fluorescent particles.

**[3b]** See response to **[3a]**.

**[3c]** How have the authors determined the intensity threshold required to call a particle fluorescent?

**[3c]** Essentially any fluorescence threshold applied to this style of instrument (including the
commercial single-particle instruments such as the UV-APS and WIBS instruments commonly
applied to atmospheric aerosol analysis) rely on what is essentially an arbitrary fluorescence
threshold limit.  WIBS users, for example, typically use a somewhat more rigorously defined
lower limit of fluorescence based on the baseline plus three sigma (standard deviations) of
observed fluorescence intensity as the lower limit. This is strongly dependent on the voltage
gain applied to the PMT detector, however, and this value is rarely monitored or reported, so
the threshold becomes somewhat arbitrary. Laboratory work is being done to understand and
find suitable calibration routines to make these procedures more standardized for such
instruments, and we plan to do similar laboratory work to develop a rigorous calibration
procedure to define thresholds for our instrument. This could be done, for example, by
measuring fluorescence spectra of a standard particle with a known fluorescence intensity. This
in itself is a very challenging task, however, because fluorescence intensity for most particles
varies as a function of age and chemical environment, not to mention particle size. So the
development of routine procedures are planned for future developments, but are excluded
from the present manuscript due to the high level of complication and complexity.

4. **[4a]** Can the authors provide the dimensions (distance to camera and angle theta for collection) of
their two instrumental set-ups along with the imaging area and pixel size of the cameras and phones
used?

**[4a]** Some of these details were introduced previously. For example, horizontal dimension of
benchtop instrument field-of-view was presented in L178. We have extended specifics of
instrument, as requested and as outlined below, and have added Table S1 to the supplemental
information, copied below:

| Camera | | | | | | | | |
|---|---|---|---|---|---|---|---|---|
| Manufacturer | Model | Type | Color / Monochrome | Detector type | Number of pixels (x $10^6$, Mp) | Pixel matrix (L x H) | Pixel size (L x H) μm | Citation |
| Canon | Powershot A2300 HD | Point-and-Shoot | Color | CCD | 15.9 | 4608 x 3456 | 1.3 x 1.3 | [1] |
| Lumenera | Infinity 2-1R | Research microscopy | Monochrome | CCD | 1.45 | 1392 x 1040 | 4.6 x 4.6 | [2] |
| Apple | iPhone 5s | Smartphone | Color | CMOS | 8.0 | 3264 x 2448 | 1.5 x 1.5 | [3] |

In addition, the following text was added or amended:
L169: "*See supplemental Table S1 for details regarding specifications of cameras discussed*
*here.*"
L172: "At the approximate angle of first order diffraction (*e.g. approx. 11º for red, 7º for blue*
*light as defined by Eq. 1*)"
L178: "field of view (of the order of *1.0* mm *wide by 0.7 mm high under 10x magnification*)"
L580: "*For standard bench-top set-up approx. distances are as follows: objective lens to grating*
*(20.5 cm), grating to camera (11.4 cm).*"
L590: "*Canon Powershot A2300 HD camera utilized offers 4608 x 3456 square pixels 1.3 μm in*
*size.*"
L625: "*As shown, field of view is a 2 mm circle. Approx. distance from objective lens to camera is*
*6 cm.*"
Additionally, the magnification of the objective lens used is an important instrumental
parameter, as was introduced to the revised text at L253 and as was discussed in response to
Point **[1b]**. 10x objective lens used for all images shown, and these details were added to each
corresponding figure caption, i.e. at Lines L584, L602, L619, and L623.
**[4b]** I believe the combination of these choices is what determines the spectral resolution achieved and
it would be nice to walk the reader through these relationships.
**[4b]** The referee is partially correct in his/her statement about the factors that determine
spectral resolution. The physical set-up does, indeed, influence the spectral resolution. The most
important physical parameters involved, however, are the dispersion angle, as defined by the grating, and the distance from the grating to the detector (camera). Because the $\Theta$ angle is fixed
for a given grating according to Equation (1), increasing this distance the between the grating
and camera results in a longer spectral swath. The longer spectral swath projected onto a CCD
with fixed number and density of pixels per unit area or distance thus provides more spectral
resolution. If the instrument is otherwise adjusted such that the image is in focus, these
parameters influence spectral resolution most directly. As discussed in response to Referee #1
under Point **[d]**, the size and homogeneity of the particle interrogated will also influence the
limit of spectral resolution.

5. In section 4, I don't follow why a 3000 k blackbody spectrum is used to approximate a theoretical
scattering curve for NaCl. Is that supposed to read 300 k? If so the same type-o occurs in the legend of
Figure 5.

This is a simple misunderstanding, and we altered the text a bit to make sure it is less likely to
happen for other readers.  The caption text of Figure 5 states: "Reference spectrum (dashed
blue) shows calculated blackbody radiator at 3000 K multiplied by CCD sensitivity curve …", and
the text (L293) states similar text.  We clarified the manuscript text as follows:

(L293) "For comparison, Figure 5 also shows the emission spectrum from a 3000 Kelvin
blackbody, *as an approximation of the emission from the heated tungsten filament source*
*used for white light,* multiplied by the theoretical sensitivity curve of the CCD used in the
monochrome camera. The theoretical  curve *represents the spectrum that the*
*CCD should detect assuming the particle does not introduce any wavelength-dependent*
*scattering features. In this case* the measured elastic scattering curve *(black line)* matches
closely *with the theoretical curve (blue, dashed line)*, suggesting that the monochrome
camera introduces very little aberration as a function of wavelength. In contrast, the color
camera shows a spectrum with pronounced peaks that are *introduced by* the different color
pixels."